# Influence of Microstructure on Tensile Properties and Fatigue Crack Propagation Behavior for Lath Martensitic Steel

**Yongjie Deng [1], Yilong Liang [1], Fei Zhao [1], Fahong Xu [2], Ming Yang [1,*] and Shaolei Long [3,*]**

[1] College of Materials Science and Metallurgical Engineering, Guizhou University, Guiyang 550001, China; qm23596399@163.com (Y.D.); fzhao@gzu.edu.cn (F.Z.)
[2] Guizhou Special Equipment Inspection and Testing Institute, Guiyang 550025, China; xfh598@163.com
[3] School of Mechanical Engineering, Guizhou Institute of Technology, Guiyang 550003, China
[*] Correspondence: myang5@gzu.edu.cn (M.Y.); 20180875@git.edu.cn (S.L.)

**Abstract:** This paper addresses the role of multilevel microstructures on the fatigue crack propagation behavior and the tensile properties of lath martensite with different substructure sizes. Microstructure characterization of the alloy was carried out by transmission electron microscopy (TEM), scanning electron microscopy (SEM), electron back-scattering diffraction (EBSD), and optical microscopy (OM). Based on the classic Hall–Petch relationship, the results of tensile tests showed that martensitic block is the effective control unit of yield strength. Furthermore, the plasticity of lath martensite is not sensitive to grain size. The tensile deformation mechanisms were also discussed. Fatigue crack propagation tests revealed that the coarse grain has a higher crack propagation threshold and lower crack propagation rate than the fine grain in lath martensitic steel. The change in the plasticity zone ahead of the crack tip leads to the transitional behavior of the fatigue crack propagation rate. When plasticity zone sizes are equal to the block size, the fatigue crack propagation reverts to a stable propagation stage. Finally, an empirical model was established to predict the fatigue crack propagation rate of the stable propagation stage based on the tensile properties of the lath martensitic steel.

**Keywords:** lath martensite; tensile properties; fatigue crack propagation rate; stable propagation modeling



## 1. Introduction

Lath martensite is a typical structure in advanced high-strength steels such as dual phase steels, low-carbon and low-alloy steels, interstitial steels, and quenching–partitioning (QP) steels. Lath martensite has excellent mechanical properties, such as high strength, ductility, and toughness, and is widely used as gear and axle material. Recent studies have been performed to investigate its phase transformation and microstructural and resultant mechanical properties [1–11]. It has been generally accepted that the refining grain can significantly improve the mechanical properties of lath martensitic steel [2,3]. In many recent studies [4–6], the electron backscattered diffraction (EBSD) technique was employed to investigate the morphology and crystallography of the lath martensite steel. Results indicated that the lath martensite had multilevel microstructures, including parent austenite grains, martensitic packets, blocks, and laths. However, there are still many controversies as to which were the effective control units between mechanical properties and multilevel microstructures. For example, Krauss and Liang et.al [7,8] revealed that the yield strength and fracture toughness of low-carbon steels with lath martensite structures increased with the increase in the packet size. On the other hand, some other researchers believed that the martensitic block was the effective grain to control static mechanical properties. Morito et al. [6] and Zhang et al. [9] tested room-temperature tensile properties and demonstrated that a typical Hall–Petch relation exists between the yield strength and the average block width. Shibata et al. conducted micro-bending tests and proved that martensitic block boundaries were the most effective barriers to dislocation sliding [10,11].

From the point of damage tolerance, fatigue crack propagation behavior is particularly important to structural components. Microstructure can also have a significant effect on fatigue crack propagation. It has been noted that grain refinement generally results in decreased resistance to fatigue crack propagation [12,13]. However, the relationship between the substructure of lath martensite and fatigue crack propagation behavior has rarely been reported for lath martensitic steel with multilevel microstructures. It is still not clear which kind of microstructure, that is grain, packet, block, or lath, is crucial to affect the crack propagation behavior. If the substructure unit of lath martensite that determines the resistance to fatigue crack propagation can be well understood, the design of lath martensite steel with superior resistance to fatigue failure may be realized.

In this work, the tensile properties and the fatigue crack propagation rate of lath martensite steel were examined. Specifically, the effects of multilevel microstructures on tensile properties and fatigue crack propagation behavior were discussed. Additionally, a fairly exact model was established in this work to describe the relationship between tensile properties and fatigue crack propagation rates in stable propagation steps when the stress ratio is 0.1.

## 2. Experimental

### 2.1. Materials and Heat Treatment Processing

The chemical composition of the steel used in this study is examined by X-ray fluorescence spectroscopy (SHIMADZU, XRF-1800, Shimadzu, Kyoto, Japan), as shown in Table 1.

**Table 1.** Chemical composition (in wt %) of the tested steel.

| C | Si | Mn | S | P | Cr | Ni | Cu | Ti |
|---|---|---|---|---|---|---|---|---|
| 0.196 | 0.229 | 0.954 | 0.0059 | 0.014 | 1.192 | 0.031 | 0.028 | 0.041 |

All the samples were austenitized at 900 °C, 1100 °C, and 1200 °C for 120 min, followed by quenching in ice salt water (5 wt %) after which they were then tempered at 200 °C for 2 h. The different austenization temperatures were selected to tailor the different martesite microstructures. These samples are named as LQ900, HQ1100, and HQ1200 samples in this work, in which LQ represents low quenching temperature and HQ represents high quenching temperature. Meanwhile, the numbers 900, 1100, and 1200 refer to the austenization temperature.

The parent austenite grain was observed using a Leica DMI5000M optical microscope (Leica company, Wetzlar, Germany). The morphologies of the packet and block were characterized by a SUPRA40 field emission scanning electron microscopy (FESEM) (ZIESS company, Oberhausen, Germany) with an electron back scattering diffraction (EBSD) detector. For EBSD measurements, the FESEM was operated at 10 kV with a step size of 0.1 μm. EBSD measurements and analysis were performed using the Azteccrystal software. The width of the lath was measured with a transmission electron microscope (TEM, FEI Talos F200X, ThermoFisher company, Stoney Creek, USA) at 300 kV. Furthermore, fracture surfaces of fatigue crack propagation samples were carefully characterized by SEM.

### 2.2. Measurement of Tensile Properties and the Fatigue Crack Propagation Rate

The mechanical properties were evaluated by uniaxial tension tests. Standard tensile samples with a gage diameter of 8 mm and length of 40 mm were tested by the Instron8501 servo-hydraulic machine with a 1 mm/s stretch rate. The yield strength (YS), ultimate tensile strength (UTS), elongation (EL), and reduction in area (RA) were determined with three individual samples.

Fatigue crack propagation experiments were performed on an MTS Model 810 servo-hydraulic machine using compact type (CT) samples with nominal dimension 62.5 mm × 60 mm × 12.5 mm machined from the heat-treated samples, as shown in Figure 1. First, CT samples were precracked in accordance with ASTM standard E647-11

and GB/T 6398-2000. After pre-cracking, measurement of the fatigue crack propagation rate was performed by increasing the $\Delta K$ via a stepwise load-shedding technique. All fatigue testing was conducted in ambient air at room temperature and a sinusoidal waveform was applied with a frequency of 10 Hz and a stress ratios (R = $\sigma_{min}/\sigma_{max}$, where $\sigma_{min}$ and $\sigma_{max}$ are the minimum and maximum stress loading, respectively) of 0.1.

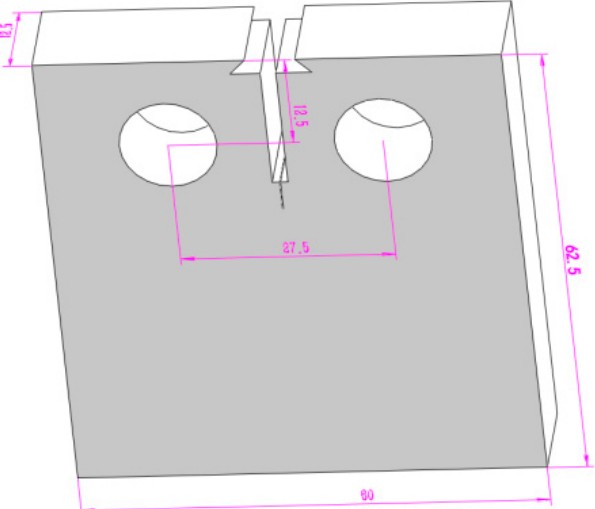

**Figure 1.** Geometry of CT sample (mm).

The compliance method was used to monitor the crack length. A crack mouth opening displacement (CMOD) gage mounted on the front face of the sample was employed to measure crack opening displacement which was then converted into crack size. The stress intensity factor range can be calculated as follows

$$\Delta K = \frac{\Delta P}{B\sqrt{W}}\frac{(2+\alpha)}{(1-\alpha)^{\frac{3}{2}}}(0.886 + 4.64\alpha - 13.32\alpha^2 + 14.72\alpha^3 - 5.6\alpha^4) \tag{1}$$

where $\alpha = a/W$ and a represents crack length; $W$, $B$, and $\Delta P$ represent the width of sample, the thickness of sample, and the applied load amplitude, respectively.

## 3. Results and Discussion

### 3.1. Microstructures

As mentioned above, the martensitic steel contained a multi-level structure like that shown schematically in Figure 2a. Figure 2b is an inverse pole figure (IPF) of the lath martensitic structure of the LQ900 sample, as quenched. Different colors represent different blocks. The parent austenite grain morphology was reconstructed by the Azteccrystal software, as shown in Figure 2c. Figure 2d is the 100, 110, and 111 actual pole figures obtained from EBSD in Figure 2b. According to the orientation information of the parent austenite grain, the standard 100, 110, and 111 pole figures of the martensite variant are shown in Figure 2e. Comparing Figure 2d with Figure 2e, it can be deduced that the tested steel maintained the exact K-S orientation relationship between parent austenite and martensitic variant.

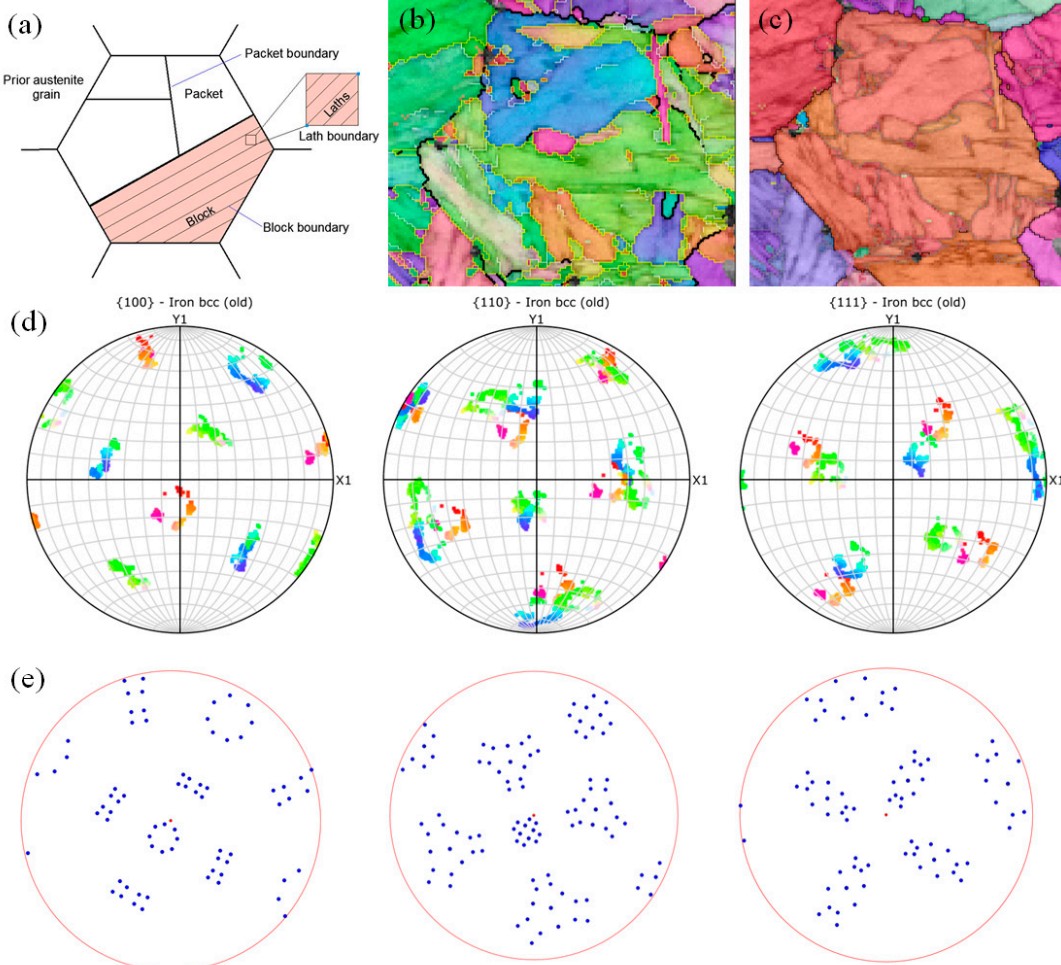

**Figure 2.** (**a**) Schematic diagram of martensite, (**b**) IPF of LQ900 sample, (**c**) the reconstructed parent austenite grain from (**b**), (**d**) actual {100} {011} and {001} pole figures obtained from EBSD in (**b**), (**e**) computed {100} {011} and {001} pole figures for 24 K-S variants for grain 1 in (**c**).

The parent austenite grain, packet, block, and lath were observed by optical microscopy (OM), SEM/EBSD, and TEM. Figure 3 shows the morphology of four substructures of all samples. The substructure size was measured by the linear intercept method using 100 grains, 100 packets, 100 blocks, and 200 laths in all samples. By means of Image-Tool software, accurate average sizes of each substructure were obtained. In Figure 3a, the diameter of the parent austenite grain increased as the austenitizing temperature increased. The grain diameters of the three samples were 17.4 μm, 70 μm, and 84.5 μm. Figure 3b gives the typical packet structure in all samples. Representative morphology and crystallographic features of the block are shown in Figure 3c. With increasing grain size, the packet and block width increased accordingly. The average packet width increased from 6.2 to 25.9 μm and the average block size increased from 2.5 to 5.9 μm as the parent grain size increased from 17.4 to 84.5 μm. Moreover, typical TEM microstructures of lath martensite are shown in Figure 3d, it is shown that lath width decreased with the grain size increased. The martensitic lath width becomes smaller at high austenitizing duo to increase the nucleation rate with of the increase in the quenching temperature [14]. The detailed data were listed in Table 2.

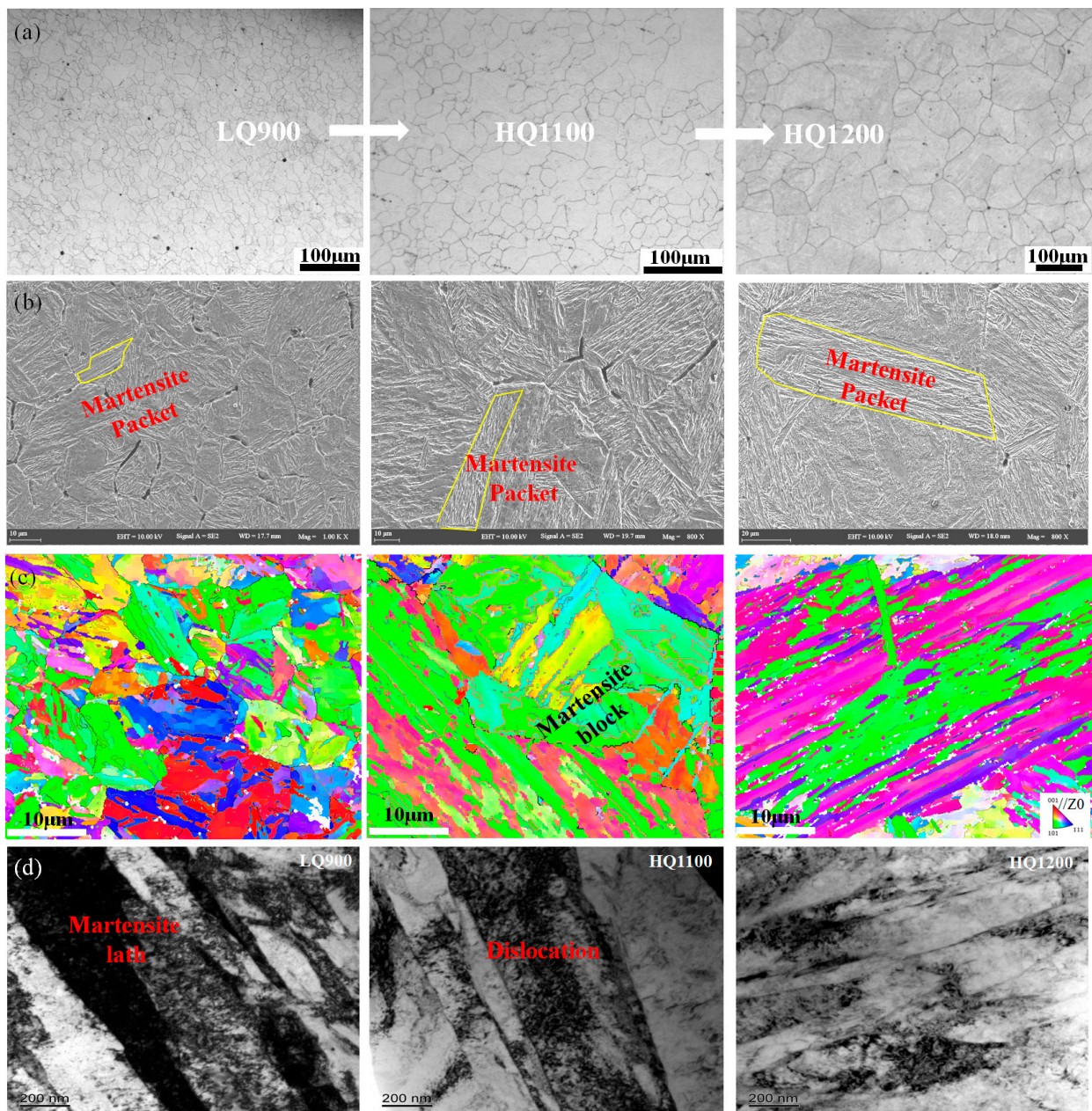

**Figure 3.** Microstructure of 20CrMnTiH steel quenched at different temperatures. (**a**) Prior austenite grain; (**b**) packet; (**c**) block; (**d**) lath.

**Table 2.** The tensile properties and microstructural dimensions obtained under different conditions.

| Samples | Grain Size ($d_g$) (μm) | Packet Size ($d_P$) (μm) | Block Size ($d_b$) (μm) | Lath Size ($d_l$) (nm) | YS (MPa) | UTS (MPa) | EL | RA |
|---|---|---|---|---|---|---|---|---|
| LQ900 | 17.4 | 6.2 | 2.5 | 270 | 1180.0 | 1375.6 | 0.14 | 0.62 |
| HQ1100 | 70.0 | 21.7 | 4.5 | 260 | 1025.3 | 1343.1 | 0.12 | 0.61 |
| HQ1200 | 84.5 | 25.9 | 5.9 | 250 | 1012.5 | 1325.4 | 0.11 | 0.59 |

*3.2. Tensile Properties*

Table 2 summarizes the mechanical properties measured on the tested steel. As expected, the LQ900 sample exhibited the highest yield strength and ultimate tensile

strength compared with the HQ1100 and HQ1200 samples. The yield strength and ultimate tensile strength of the lath martensitic steel decreased when the quenching temperature increased from 900 °C to 1200 °C. It may be resulted from the finer grain size of the LQ900 sample. The coarsened microstructure in the HQ1100 and HQ1200 samples resulted in a lower yield strength and the ultimate tensile strength.

As mentioned previously [5,6], martensite is subdivided by packet, block, and lath; every substructure's boundary plays an important role in impeding the motion of dislocations. To find the substructure controlling the yield strength, the classic Hall–Petch formula was employed in the present work. Figure 4 shows that the prior austenite's grain size, the packet size, and the block size have excellent linear relationships with yield strength. Since the block unit was the smallest substructure in the three microstructures, it can be considered as an effective control unit for yield strength in lath martensite. In contrast, lath size displayed a nonlinear relationship with yield strength. That is because the lath boundary was a low angle boundary, which cannot impede the motion of dislocation [15,16].

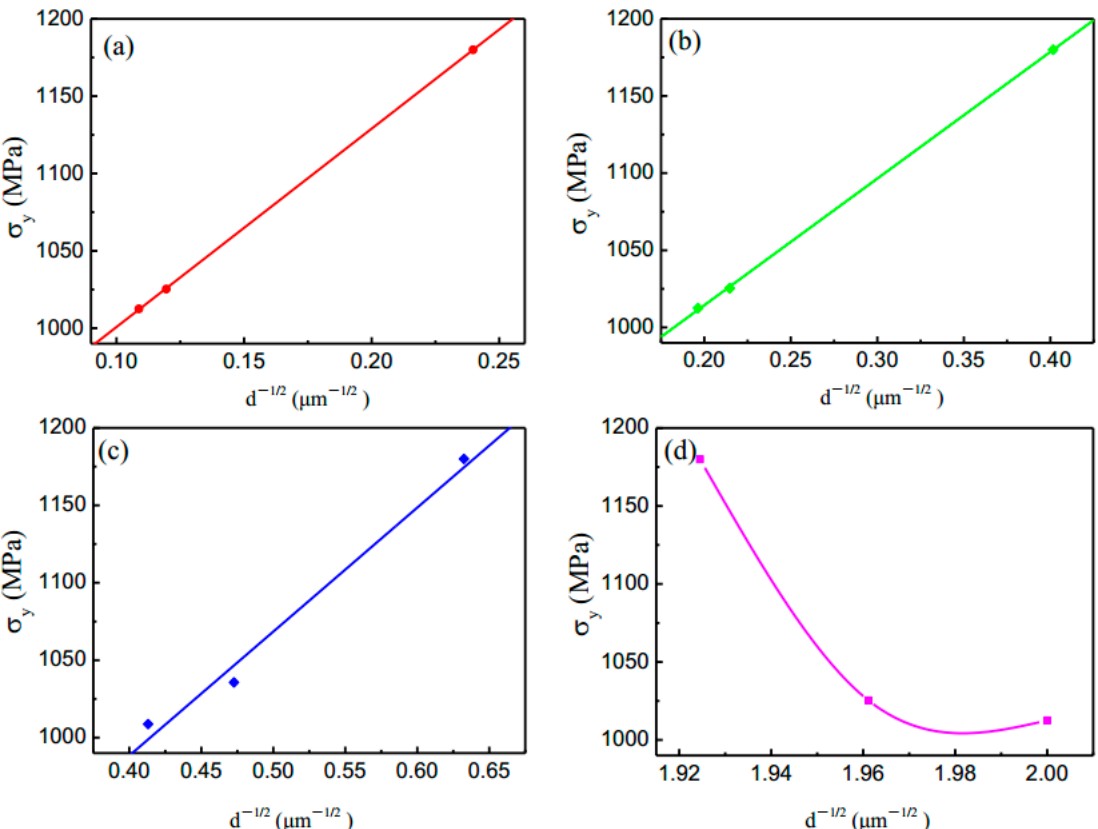

**Figure 4.** The variation trend of yield strength and parent austenite grain size (**a**), packet size (**b**), block width (**c**), and lath width (**d**) in the tested steel.

The tensile properties demonstrated the elongation and reduction in area of all samples were in the range of 11~13% and 59~62%, respectively. The elongation and reduction in area of the LQ900 sample were slightly higher than those of the HQ1100 and HQ1200 samples. It is suggested that the ductility of the alloy is not sensitive to grain size. Generally, the plastic deformation ability of metal crystals can be judged by the Schmid factor. When the Schmid factor is large, the stress level required for the initiation of dislocation slip in the crystal is lower and the plasticity of the crystal is better. For the martensitic structure with BCC structure, the dislocation slip systems include {110}[111] and {112}[111] slip systems. The distribution map of the Schmid factor of LQ900, HQ1100, and HQ1200 samples can be obtained by the EBSD technique, as shown in Figure 5. It can be seen that {110}[111] and {112}[111] slip systems activate in all the samples. The average Schmid factors of the three

samples were 0.4677, 0.4606, and 0.4753, respectively. It means that the ability of dislocation slip activation in these samples is similar and the change in grain size has little effect on the activation of dislocation slip.

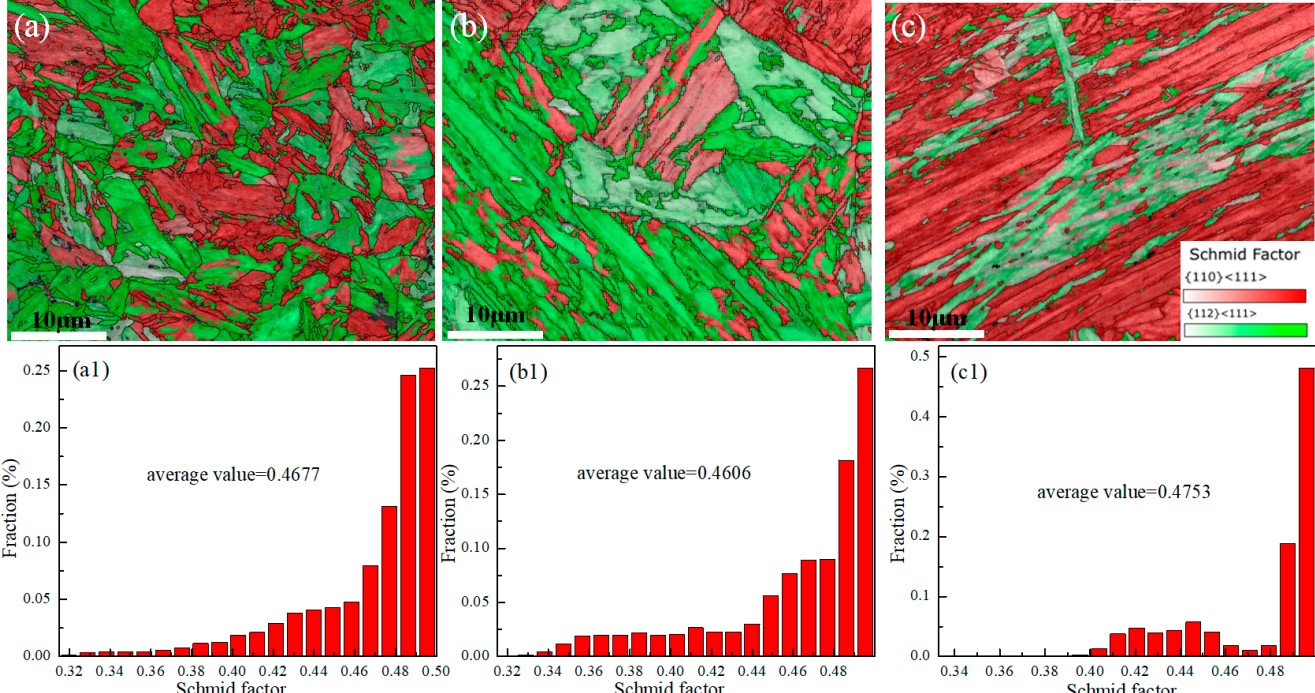

**Figure 5.** The Schmid factor (**a–c**) and distribution maps (**a1–c1**) of LQ900, HQ1100, and HQ1200 samples.

It is well known that fine grains have better plastic deformation ability. However, compared with the LQ900 sample, there were only a 2% and 3% reduction in elongation and reduction in area in the HQ1200 sample. In this work, the ductility was not significantly different in the coarse and refined grains of lath martensite, which is fully different from other steels such as medium-carbon steel and high-carbon steel.

For a better understanding of the detailed tensile deformation behavior, the tensile fracture profile was obtained by sectioning the HQ1200 sample along the middle plane parallel to the tensile axis observed by EBSD. Orientation mapping (Figure 6) of the tensile fracture surface showed that the single color of the block changed to different grayscales. Accordingly, the misorientation of the martensite block changed from a single orientation to several orientations. In addition, the inverse pole figures in Figure 6 presented the motion of the block.

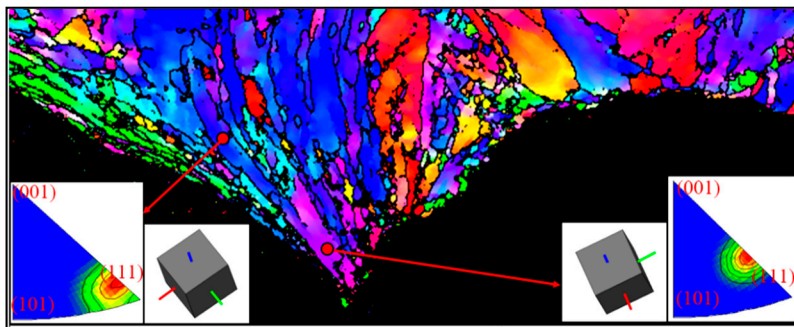

**Figure 6.** Orientation mapping of the tensile fracture profile of the HQ1200 sample.

To further understand the contribution of rotation and interface sliding to plastic deformation, the average misorientation map was obtained as shown in Figure 7. Figure 7a,b shows the average misorientation of a block of testing alloys quenched at 900 and 1200 °C, respectively. Figure 7c,d displays the corresponding statistical data. It can be found that the HQ1200 sample with more coarse grains had a bigger average misorientation value. From the results obtained by EBSD analysis, it can be concluded that the orientation transformation behavior results from the boundary sliding and rotation. During tension deformation, most of the applied strains have been accommodated by the sliding and motion of substructure boundaries, such as rotation and bending. Therefore, the change in plasticity can be explained using the combined effect of the dislocation sliding mechanism and the grain boundary deformation mechanism. The dislocation sliding mechanism predominates in most metal materials. Plastic deformation mainly occurs by lattice dislocations in individual grains. However, when grains are refined to the submicrometer or nanometer scale, plastic deformation is mainly attributed to grain boundary sliding and the grain boundary sliding mechanism becomes the dominant mechanism [17]. For lath martensite, lath units are virtually single crystals with thicknesses of approximately hundreds of nanometers [18]. From a specific viewpoint, the multilevel lath martensitic could be considered an ultrafine grain microstructure. Consequently, dislocation activity within laths and significant plastic accommodation in the vicinity of lath boundaries, block boundaries, packet boundaries, and grain boundaries improve the plasticity of lath martensite with a coarse grain size.

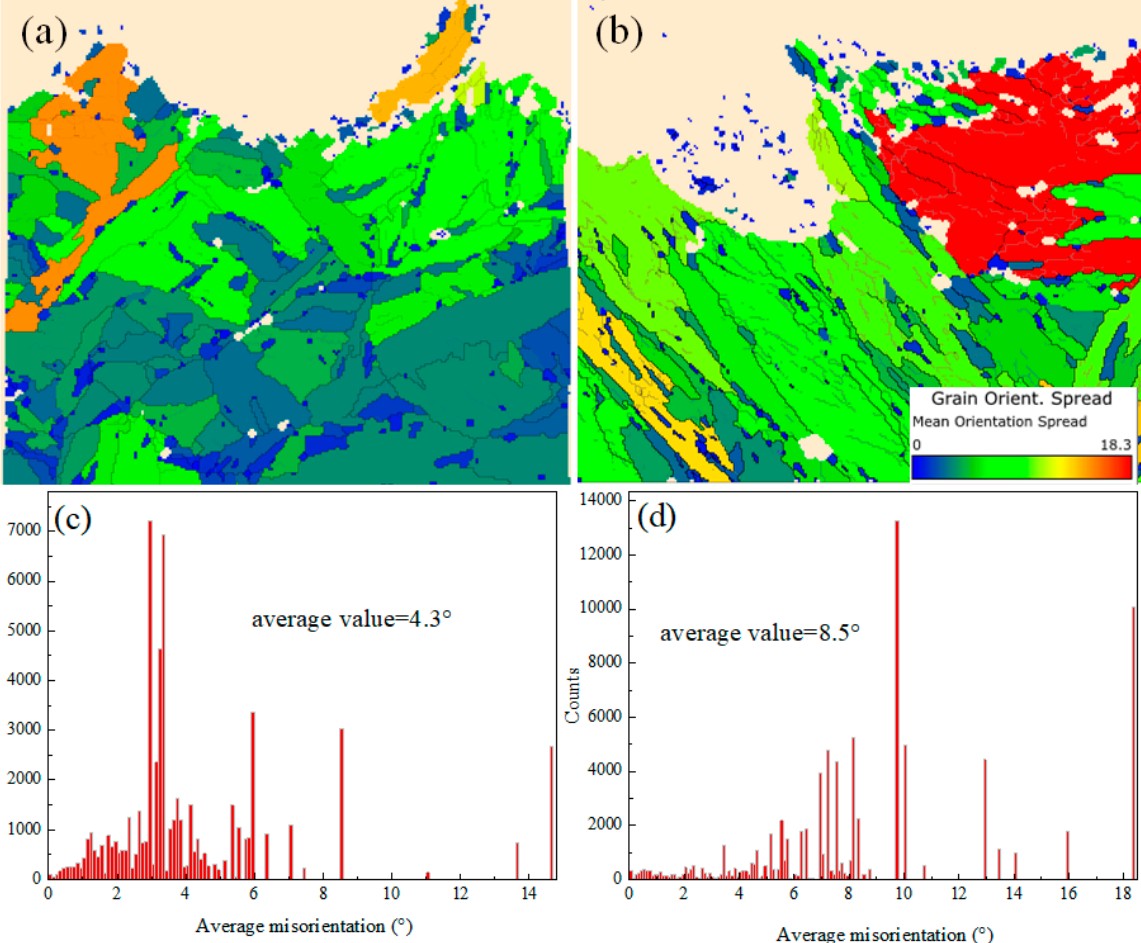

**Figure 7.** The average misorientation (**a**,**b**) and distribution maps (**c**,**d**): (**a**,**c**) are the LQ900 sample and (**b**,**d**) are the HQ1200 sample.

The above analysis was also supported by recent studies. Morsdorf et al. [19] investigated the role of martensite substructure boundaries and pointed out that high shear stresses for nearly 45 orientations enable the most pronounced boundary plasticity. Furthermore, a more microscopic analysis was carried out by Du et al. [20]. Through uniaxial microtensile tests, they concluded that the sliding of boundaries can be activated in all types of boundaries: lath, subblock, and block boundaries. Plastic deformation of lath martensite resulted from the competition of dislocation slip and boundary sliding during plastic deformation.

This fracture surface coincided well with our experimental results on the prominent plasticity of lath martensite. Fractographies of the LQ900, HQ1100, and HQ1200 samples are shown in Figure 8. At the low level of magnification, the fracture surfaces of these samples had a similar typical cup–cone fracture appearance (cone portion), as shown in Figure 8a. The fracture morphologies of all three samples revealed ductile fracture modes. In addition, the tensile fracture surfaces of all samples were obviously divided into two zones: a fibrous zone and a shear lip zone. The central region of the surfaces was the fibrous zone. The outside of the surface were shear lip zones which displayed a strong shear character. It means that cracks nucleate and grow during the process of tensile fracture and consume most of the energy. Moreover, it was found from the low-magnification photography that the areas of the fibrous zone in all samples were almost the same. At higher magnification (1000× or higher), many dimples can be observed in the fibrous zone. These deep dimples indicated that the tensile fracture mechanism in lath martensite was micro-void coalescence. That is, the dimples were produced by voids nucleating ahead of the principal crack, which has a blunted tip because of the plasticity of the material. Therefore, the number and depth of dimples are associated with the material's plasticity. As shown in Figure 8b, the fracture mode of all these samples was also mainly the ductile fracture mechanism. Further careful observation and analysis showed that, unlike the LQ900 sample, the HQ1100 and HQ1200 samples contained some microcracks and tearing ridges in the fibrous zones. Meanwhile, small quasi-cleavage steps at the bottom of tearing ridges were observed in the high-magnification image. This is the reason why the HQ1100 and HQ1200 samples have slightly lower elongation and reduction in area.

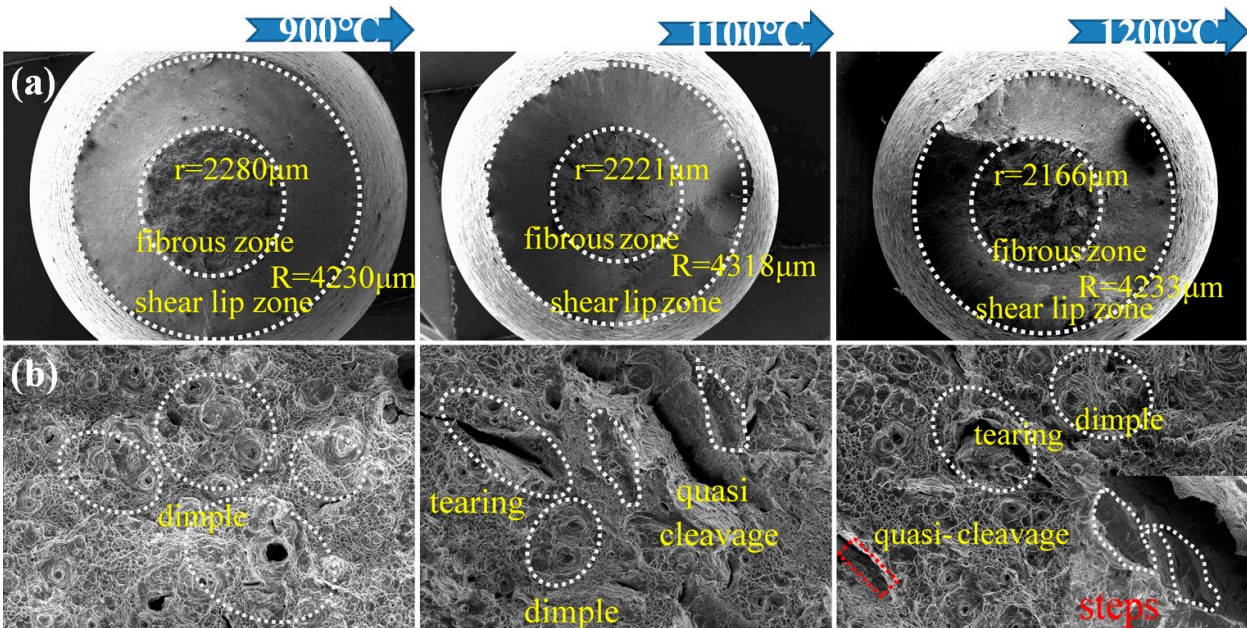

**Figure 8.** Fractographies of the LQ900, HQ1100, and HQ1200 samples after tensile deformation: (**a**) microfracture and (**b**) high magnification of the center zone in fracture surfaces.

### 3.3. Fatigue Crack Propagation Behavior

Figure 9 depicts the crack length increment ($da/dN$) versus stress intensity factor ($\Delta K$) curves of the LQ900, HQ1000, and HQ1200 samples. The fatigue crack propagation curves essentially consisted of three regions: a threshold region (region I), a stable propagation region (region II), and an unstable propagation region (region III). In the threshold region, there were two characteristic values: the threshold cyclic stress intensity factor ($\Delta K_{th}$) and the stress intensity factor transitional value ($\Delta K_T$). The stress intensity factor transitional value ($\Delta K_T$) was displayed in Table 3.

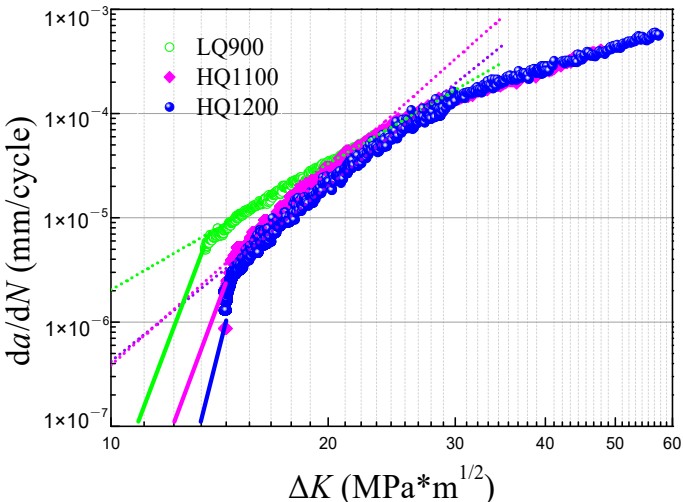

**Figure 9.** FCGR curves and transition points for the tested steel.

**Table 3.** The characteristic size of the transitional point.

| Samples | LQ900 | HQ1100 | HQ1200 |
|---|---|---|---|
| Transition point (Mpa$\sqrt{}$m) | 12 | 14.1 | 14.5 |

In the stable propagation region, $\Delta K$ and $da/dN$ are displayed as nearly linear on a $\log K$–$\log(da/dN)$ scale. The linear relationship can be described with the Paris–Erdogan power model [21]; the Paris constants are listed in Table 4.

$$da/dN = C(\Delta K)^m \qquad (2)$$

Here, a is the crack length, $N$ is the number of cycles, $\Delta K$ is the cyclic stress intensity factor, and $C$ and m are empirical constants that depend on the material, environment, and test conditions.

**Table 4.** Fitting results of the Paris–Erdogan power model in tested samples.

| Samples | C | m | Correlative Coefficient |
|---|---|---|---|
| LQ900 | $1.40 \times 10^{-10}$ | 4.13 | 099.5% |
| HQ1100 | $4.55 \times 10^{-13}$ | 6.04 | 97.6% |
| HQ1200 | $2.16 \times 10^{-12}$ | 5.36 | 98.0% |

In the crack unstable propagation stage, the crack propagation rates displayed a significant increase, indicating that the accelerated crack propagation rate was associated with the start of final rupture.

Specifically, the HQ1200 sample exhibited the lowest crack propagation rate and highest threshold value compared with the HQ1100 and LQ900 samples. Figure 9 shows that in the near-threshold region, the fatigue crack propagation curve had a transition at $\Delta K$ = 12 MPa$\sqrt{m}$ for the LQ900 sample, $\Delta K$ = 14 MPa$\sqrt{m}$ for the HQ1100 sample, and $\Delta K$ = 14.5 MPa$\sqrt{m}$ for the HQ1200 sample.

According to the micromechanics proposed by McClintock et al. [22], fatigue cracks cease to advance when the plastic zone size of the crack tip is equal to a certain characteristic microstructural size. Hence, the transition behavior is correlated to certain microstructural sizes. Using Equation (3) [23], the cyclic plastic zone at the transition point can be calculated.

$$r_c = \frac{1}{\alpha\pi}\left(\frac{\Delta K}{2\sigma_y}\right)^2 \tag{3}$$

Here, $\alpha$ is equal to three for the plane strain condition and $\sigma_y$ is the yield strength. Comparing the cyclic plastic size to lath martensite substructures, it is interesting to note that the cyclic plastic radius was almost equal to the block width. Therefore, the martensitic block is considered a characteristic microstructure that controls the transition behavior of the fatigue crack propagation rate in the threshold region.

Additionally, the crack propagation threshold values were obtained by extending the curves to the x-axis at $da/dN$ = $10^{-7}$ mm/cycle, which were approximately 10 MPa$\sqrt{m}$, 11 MPa$\sqrt{m}$, and 12 MPa$\sqrt{m}$ for the LQ900, HQ1100, and HQ1200 samples, respectively. The results demonstrated that the higher austenitic temperature led to a pronounced improvement in the threshold value and the resistance of crack propagation. The reasons for these differences were discussed as follows. The effect of the macroscopic crack path was first considered. Figure 10a displays the fatigue crack propagation paths of the LQ900 and HQ1200 samples. It can be found that the LQ900 sample displayed a relatively smooth and straight crack propagation path with no obvious deflection. However, crack deflections were more prominent in the HQ1200 sample. In Figure 10b,c, it can be seen that the crack deflected at the grain boundary and pocket boundary and the crack propagation paths exhibit a tortuous zigzag within the packet. The crack path images clearly showed that the fatigue crack propagation behavior was strongly influenced by the multilevel substructure of martensite. It is well known that parent grain size has a positive correlation with the length of dislocation sliding [24]. Thus, the coarse grains accelerated slip reversibility, resulting in a decrease in damage accumulation of crack tips and they will consume more energy for crack propagation [25]. Secondly, the difference in crack deflections illustrated that the fracture surface roughness was controlled by the microstructure. The crack surface roughness of the LQ900 and HQ1200 samples was measured by an OLS4500 laser scanning confocal microscope produced by Olympus, as shown in Figure 11. In Figure 11a, the HQ1200 sample showed relatively larger local roughness than the LQ900 sample. To provide a quantitative representation, the profile lines of the LQ900 and HQ1200 samples along the sample centerline are shown in Figure 11b. It is obvious that numerous crack deflections occurring in coarse-grained samples made the fracture surfaces rougher, resulting in a higher degree of mismatch between the matching cracked surfaces. These mismatched surfaces reduced the crack propagation driving force and thus slowed down the crack propagation rate, generating a crack closure effect.

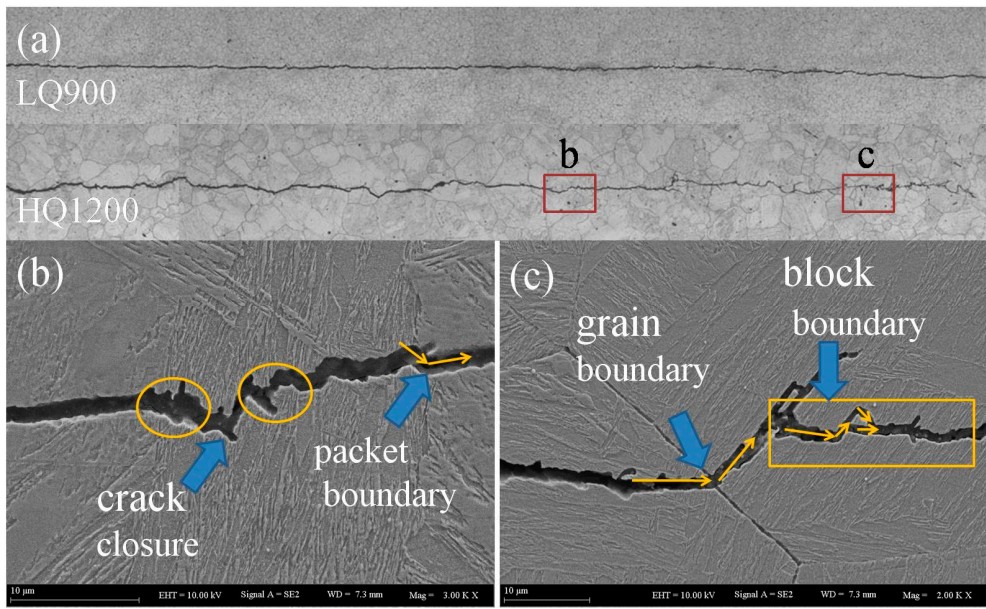

**Figure 10.** (**a**) Illustrations of the fatigue crack propagation paths of LQ900 and HQ1200; (**b**) crack closure; and (**c**) crack deflection.

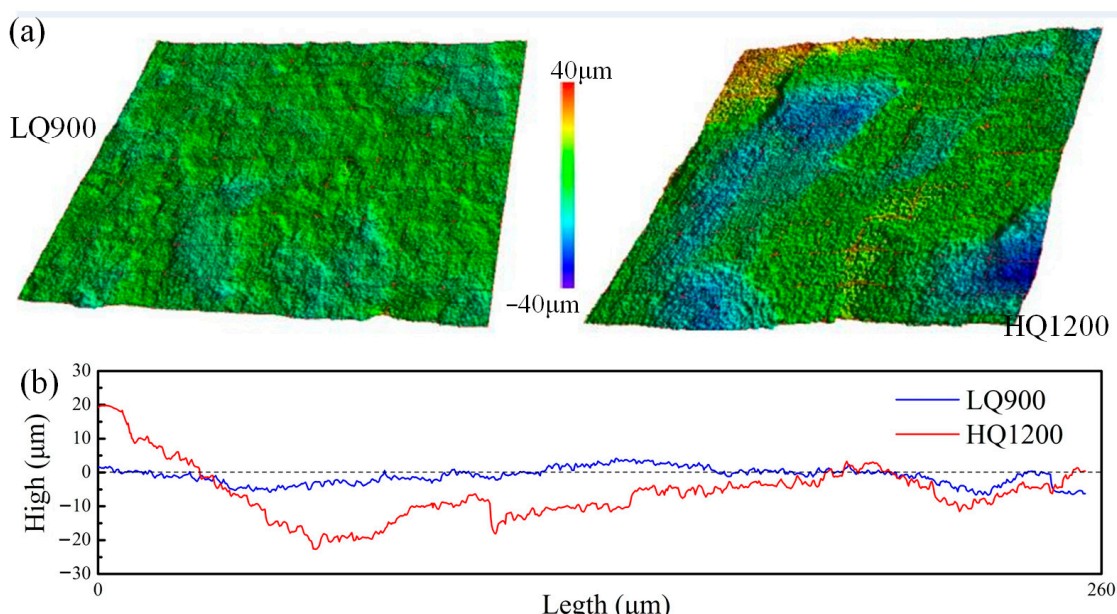

**Figure 11.** (**a**) Three-dimensional metallographic photographs and (**b**) surface line profiles of fatigue fracture surface morphology of LQ900 and HQ1200 samples.

### 3.4. The Model of the Fatigue Crack Propagation Rate in the Stable Propagation Stage

The tensile properties and the fatigue crack propagation rate are significant variables for engineering materials. Generally, the measurement of the fatigue crack propagation rate is more technically complicated, more expensive, and more time consuming. Thus, establishing the fundamental relationship between the tensile properties and the fatigue crack propagation rate is a valuable endeavor. Consequently, the crack propagation rate can be predicted by means of the tensile test data with this relationship.

Using the predicted model of the threshold value ($\Delta K_{th}$) proposed by Fleck et al. [26], it can be found that $K_{IC}$, $\Delta K_{th}$, and the Paris parameter $m$ maintained the following relationship:

$$\log\left(\frac{K_{IC}}{\Delta K_{th}}\right) = \frac{4}{m} \text{ Or } \log\left(\frac{K_{IC}}{\Delta K_{th}}\right) \propto \frac{1}{m} \tag{4}$$

where $K_{IC}$ is the static plane strain fracture toughness, $\Delta K_{th}$ is the threshold stress intensity factor range, and m is the Paris parameter. On the other hand, previous studies suggested that both $K_{IC}$ and $\Delta K_{th}$ can be calculated in terms of tensile properties such as yield strength (YS), elongation (EL), and reduction in area (RA). Thus, the relationship between tensile properties and the crack propagation rate can be established in this work based on the model mentioned above.

Yu et al. [27] provided a simple analytical model to calculate $\Delta K_{th}$ with fatigue crack propagation at R = 0, as shown in Equation (5):

$$\Delta K_{th} = E\varepsilon_f\sqrt{2\pi\rho_c} \tag{5}$$

where $E$ is the elastic module, $\varepsilon_f$ is the true fracture strain, and $\rho_c$ is the critical root radius for fatigue crack passivation zone. Hahn and Rosenfield [28], Zheng et al. [29], and Richards et al. [30] investigated the relationship between $K_{IC}$ and the uniaxial tensile properties of materials. They established some empirical models. All the models were found to obey the following equation:

$$K_{IC} \propto \sqrt{dE\varepsilon_f\sigma_s} \tag{6}$$

where $d$ is a characteristic size related to the microstructure, $E$ is the elastic model, $\varepsilon_f$ is the true fracture strain, and $\sigma_s$ is the yield strength. Upon substituting Equations (5) and (6) into Equation (4), Equation (4) can be expressed as:

$$\log\left(\frac{K_{IC}}{\Delta K_{th}}\right) \propto \log(\frac{\sqrt{dE\varepsilon_f\sigma_s}}{E\varepsilon_f\sqrt{2\pi\rho_c}}) \propto \frac{1}{m} \tag{7}$$

Notably, $\rho_c$ and d are constant for the same microstructure. Thus, Equation (7) can be described as the following equation:

$$\log(\frac{\sigma_s}{\varepsilon_f}) \propto \frac{1}{m} \tag{8}$$

In Equation (8), the value of $\varepsilon f$ can be given from Equation (9) [31].

$$\varepsilon_f = \ln(\frac{1}{1-RA}) \tag{9}$$

where $\varepsilon_f$ and $RA$ are the true fracture strain and reduction in area, respectively.

In other words, the equation indicates that $1/m$ has a strong correlation with $\log(\sigma_s/\varepsilon_f)$. The plot of $\log(\sigma_s/\varepsilon_f)$ versus $1/m$ for lath martensite is given in Figure 12a. The data show an excellent linear correlation. The function fitting result can be obtained from the slope and intercept and can be described as:

$$\lg(\frac{\sigma_s}{\varepsilon_f}) = 2.952 + 0.562(\frac{1}{m}) \tag{10}$$

which yields

$$m = \frac{0.562}{\lg(\frac{\sigma_s}{\varepsilon_f}) - 2.952} \tag{11}$$

For lath martensitic steel, the relationship between log$C$ and $m$ is shown in Figure 12b. The result also illustrated that the relationship of these two parameters also obey good linear correlation, as shown in Equation (12).

$$\log C = -4.435 - 1.324m \tag{12}$$

which yields

$$C = 10^{-4.435 - 1.324m} \tag{13}$$

Substituting Equations (11) and (13) into the Paris model (Equation (2)) yields

$$da/dN = C(\Delta K)^m = 10^{-4.435 - 1.324m}(\Delta K)^m \tag{14}$$

$$da/dN = 10^{-3.1238 - 1.752\frac{0.562}{\lg(\frac{\sigma_s}{\varepsilon_f}) - 2.952}}(\Delta K)^{\frac{0.562}{\lg(\frac{\sigma_s}{\varepsilon_f}) - 2.952}} \tag{15}$$

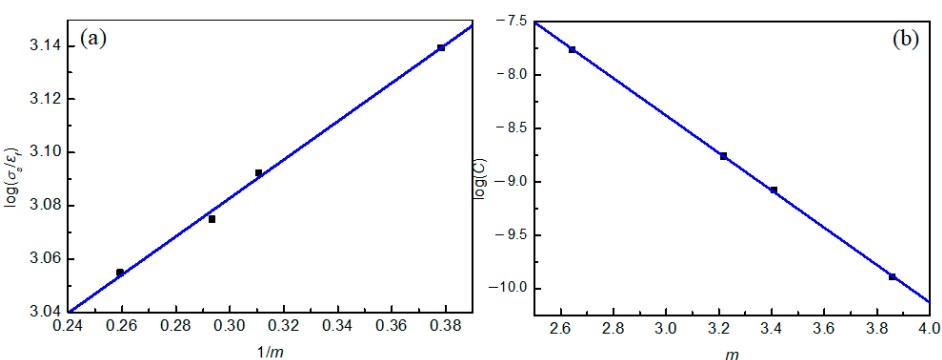

**Figure 12.** The linear relationships between (**a**) $\lg(\sigma_s/\varepsilon_f)$ and $1/m$ (**b**) $m$ and log$C$.

A comparison of the experimental results and the predicted crack propagation rate is shown in Figure 13. The predicted values were in good agreement with the experimental values. Thus, when the yield strength and the reduction in area were obtained by a standardized tensile test, based on Equation (15), the fatigue crack propagation rate of lath martensitic steel in the stable propagation stage at R = 0.1 can be approximately evaluated.

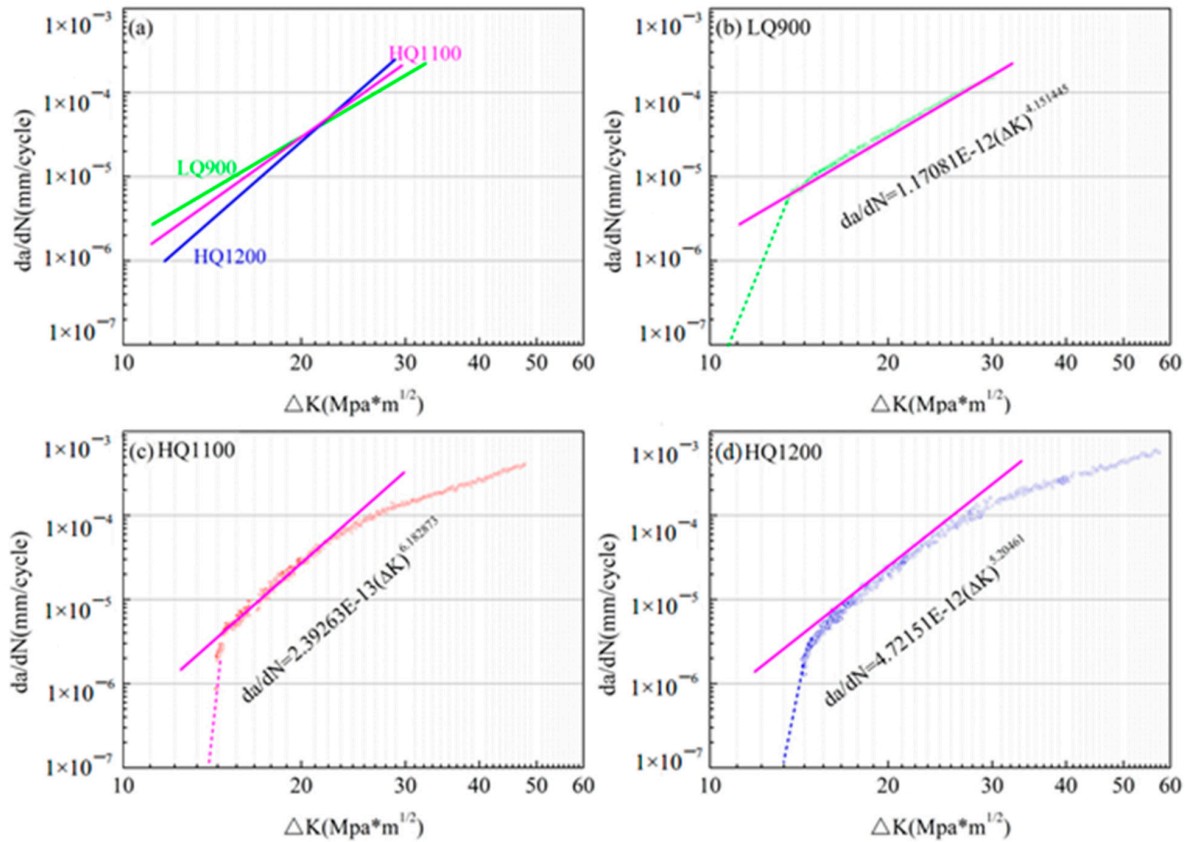

**Figure 13.** Fatigue crack propagation rate: (**a**) the predicted data, the predicted data and the experimental results for (**b**) LQ900, (**c**) HQ1100, and (**d**) HQ1200 samples.

## 4. Conclusions

The tensile properties and fatigue crack propagation behavior of lath martensite with different parent grain sizes were presented. The main conclusions are the following:

(1)  Based on the classic Hall–Petch formula, the block width is considered as the effective grain size to control the yield strength of lath martensite;

(2)  For lath martensitic steel with different substructure sizes, neither elongation nor reduction in area resulted in no noticeable reduction. It reveals that boundary sliding is an important deformation mechanism for lath martensite steel and has excellent plasticity benefits, from the crystallographic slip in the lath and plastic accommodation in the vicinity of substructure boundaries;

(3)  For lath martensitic steel, coarse grain structures have better fatigue crack propagation resistance than finer grains. The improved resistance comes from the boundary resistance of multilevel microstructures and closure effects. The transition point in the fatigue crack propagation rate curve depends on the block size;

(4)  Based on the tensile properties, an empirical model was established to evaluate the fatigue crack propagation rate of the stable propagation region for lath martensitic steel; the predicted data are in good agreement with the experimental results.

**Author Contributions:** Y.D.: Conceptualization, Data curation, Formal analysis, Methodology, Writing—original draft, Software. S.L.: Validation, Methodology. M.Y.: Resources, Methodology, Project administration, Supervision, Writing—review and editing. F.Z.: Validation, Visualization. F.X.: Validation. Y.L.: Validation. All authors have read and agreed to the published version of the manuscript.

**Funding:** The authors acknowledge support from the Natural Science Foundation of Guizhou Province (Grant No. [2020]1Z046, NO; ZK[2022] 023; [2020]1Y199).

**Data Availability Statement:** Data will be made available on request.

**Conflicts of Interest:** The authors declare no conflict of interest.

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
