# Peer review of "Influence of Microstructure on Tensile Properties and Fatigue Crack Propagation Behavior for Lath Martensitic Steel"

_crystals, doi:10.3390/cryst13091392_

Round 1

Reviewer 1 Report

This paper studied the effect of substructure on fatigue crack characteristics in lath martensitic steel through systematic microstructural analysis and mechanical property evaluation analysis.

A few things below need to be corrected.

 1)       It is necessary to add an explanation for the reason why the lath width becomes smaller at high austenitizing temperature by referring to other research papers.

 2)       Page 6, line 16-164: Add definitive reference citation that lath martensite does not effectively impede dislocation migration. Because this is not fully explained in reference 14.

3)       Figure 5 and it’s description(The schmid factor (a~c) and distribution maps (c~d) of LQ900, HQ1100 and HQ1200 samples) do not match.

4)       Sentence corrections such as unnatural English expressions and grammar corrections are needed.

Moderate editing of English language required

Author Response

  1. It is necessary to add an explanation for the reason why the lath width becomes smaller at high austenitizing temperature by referring to other research papers.

Response: We appreciate the valuable comments from you. The reference has been cited to explain the smaller martensitic lath with the increasing of quenching temperature.

[14]Shao-lei Long, Yi-long Liang, Yun Jiang, Ming Yang, et.al. Effect of quenching temperature on martensite multi-level microstructures and properties of strength and toughness in 20CrNi2Mo steel, Materials Science & Engineering A, 2016, 676: 38-47

  1. Page 6, line 16-164: Add definitive reference citation that lath martensite does not effectively impede dislocation migration. Because this is not fully explained in reference 14.

Response: We sincerely thank the reviewer for careful reading. The reference 16 was added to further explain the opinion that martensite lath does not effectively impede dislocation migration.

[16]Mayama,Tsuyoshi,Takashima,et al. Anisotropy of strength and plasticity in lath martensite steel[J].Materials Science & Engineering, A. 2016,674: 104-116

  1. Figure 5 and it’s description(The schmid factor (a~c) and distribution maps (c~d) of LQ900, HQ1100 and HQ1200 samples) do not match.

Response: We sincerely apologize for this oversight. We have corrected the errors accordingly.

  1. Sentence corrections such as unnatural English expressions and grammar corrections are needed.

Response: We sincerely thank the reviewer for careful reading. The manuscript is checked carefully, with particular attention to avoiding English mistakes. And it has been proofread by someone skilled in English from a certified editing agency.

Reviewer 2 Report

Article "Effect of substructure on tensile properties and fatigue crack growth behavior for lath martensitic steel" by Yongjie Deng, Yilong Liang, Fei Zhao, FaHong Xu, Ming Yang and ShaoLei Long.

The paper submitted for review is interesting from the point of view of materials science. The article touches upon an important topic - the study of the structural elements sizes impact (in this case, substructure elements sizes) on the mechanical properties of metallic materials. The issue is important because, unfortunately, many existing models of crack formation and crack propagation in steels do not take into account the influence of the structure, or this account is carried out very superficially. In addition, despite the fact that the main patterns of the structural elements sizes impact on mechanical properties have long been studied, modern studies show more and more specific cases when deviations from these laws can be observed, that is associated with the structural features of the material. In this regard, the presented paper is of scientific interest.

The paper as a whole is constructed well; the main idea and conclusions are stated correctly. There are a number of comments to the article, which should be eliminated before the publication.

1. Some sentences in the abstract are formulated poorly in terms of grammar. It is necessary to check the abstract, and the article as a whole should be checked by a native speaker. Also in the abstract, at the first reading of the article, it is not clear what “block” is referred to (line 14).

2. The references list is approximately half made up of papers older than 20 years. With all due respect to the "classical" works, it is necessary to supplement the list of references with a number of modern researches on the paper subject.

3. Please check if the expression for “load ratio R” (lines 100-102) is correct.

4. Expression (1) is given with specific values of the coefficients. In this case, it is necessary to indicate in which units it is necessary to substitute the physical quantities included in this expression.

5. 900LQ must be replaced by LQ900 (line 115).

6. In Figure 2, the figure indicates the designation of the set of crystallographic planes {XXX}, and in the caption - the designation of the single crystallographic plane (XXX). It is necessary to make the same designations in the figure and in the caption.

7. The designation of the specimen cross-section reduction must be made the same in the article - RA or Z.

8. Figure 4d, showing the dependence of the yield strength on the size of the lath, is quite controversial. The difference in lath size between the three samples is too small to plot. The error in determining the lath size by metallographic methods can exceed the difference in lath sizes for different samples.

At the same time, it is necessary to mention the following fact. In the literature, there are several cases described when for materials with an ultrafine structure (the size of the structural element is less than 1 μm), deviations from the Hall-Petch law are observed (a decrease in the value of the yield strength with decreasing grain size). I believe that in order to draw a correct conclusion about the effect of the lath size on the yield strength, a large measurement statistics is needed.

9. In general, the article should be carefully read before publication to correct minor inaccuracies.

 ___

General conclusion: the paper is of scientific value, interesting to read and may have practical output. I believe that after the elimination of the remarks, the article can be published in "Crystals".

Author Response

  1. Some sentences in the abstract are formulated poorly in terms of grammar. It is necessary to check the abstract, and the article as a whole should be checked by a native speaker. Also in the abstract, at the first reading of the article, it is not clear what “block” is referred to (line 14).

Response: We sincerely thank the reviewer for careful reading. The manuscript is checked carefully, with particular attention to avoiding English mistakes. And it has been proofread by someone skilled in English from a certified editing agency. In the abstract, the block has been replaced with martensitic block, and the definition of martensitic block was explained in the introduction.

  1. The references list is approximately half made up of papers older than 20 years. With all due respect to the "classical" works, it is necessary to supplement the list of references with a number of modern researches on the paper subject.

Response: Thank you very much for your valuable suggestion. The references have been updated in the revised manuscript.

  1. Please check if the expression for “load ratio R” (lines 100-102) is correct.

Response: The word load has been replaced with loading.

  1. Expression (1) is given with specific values of the coefficients. In this case, it is necessary to indicate in which units it is necessary to substitute the physical quantities included in this expression.

Response: We sincerely thank the reviewer for careful reading. In expression (1), α=a/W,and a, B, W, and ∆P are crack length, the thickness of specimen, the width of specimen and the applied load amplitude, respectively. In fact, the coefficient α is variant due to the variation of crack length in fatigue crack growth test.

     (1)

  1. 900LQ must be replaced by LQ900 (line 115).

Response: The error has been revised.

  1. In Figure 2, the figure indicates the designation of the set of crystallographic planes {XXX}, and in the caption - the designation of the single crystallographic plane (XXX). It is necessary to make the same designations in the figure and in the caption.

Response: These errors have been revised.

  1. The designation of the specimen cross-section reduction must be made the same in the article - RA or Z.

Response: The Z in table 2 has been replaced with RA.

  1. Figure 4d, showing the dependence of the yield strength on the size of the lath, is quite controversial. The difference in lath size between the three samples is too small to plot. The error in determining the lath size by metallographic methods can exceed the difference in lath sizes for different samples.

At the same time, it is necessary to mention the following fact. In the literature, there are several cases described when for materials with an ultrafine structure (the size of the structural element is less than 1 μm), deviations from the Hall-Petch law are observed (a decrease in the value of the yield strength with decreasing grain size). I believe that in order to draw a correct conclusion about the effect of the lath size on the yield strength, a large measurement statistics is needed.

Response: According to classical Hall-Petch law, the yield strength is proportional to d-1/2. Figure4 provide information regarding to the effectively control unit for strength by the classical formula of Hall–Petch. The results indicate that the lath size does not follow the Hall–Petch relationship. In fact, lath size remains almost unchanged with quenching temperature increasing. The testing results were confirmed with a large measurement statistics.

  1. In general, the article should be carefully read before publication to correct minor inaccuracies.

Response: We sincerely apologize for this oversight. We have carefully checked the manuscript and corrected these errors accordingly.  

Reviewer 3 Report

Dear Authors,

Thank you for submitting your paper to Crystals. In your article, you try to explain tensile properties and fatigue crack behavior by the analysis of the microstructure. It is a very interesting and very hard-to-catch subject. Only recently I had read a PhD thesis devoted to microstructure and fracture toughness. Your paper is clear but as always some editorial problems appear.

In lines 125, 153, and 331("or" in eq. 3) there is a problem with font size.

In lines 158, 214, 215, and 282  you left crossed-out text.

In lines 172, 176, and 178 there should be "Schmid".

In line 332 you name Kic as static strain fracture toughness. I think you missed "plane strain" in this name.

Il line 365 Eq. 13 is hardly readable. You can cut the beginning that is taken from the preceding line.

In line 106 you stated that you computed COD using CMOD. Did you assume that the edges of the crack are curvilinear?

When I saw Fig.8 I wondered if you intend to take into account local discrepancies in the state of stress which can lead to these tearing-like voids?

Did you try to correlate the fracture toughness with the surface parameter? Maybe the surface roughness parameter will be good. Years ago there were attempts to use the fractal dimension of the profile but these were abandoned.

Best regards,

Author Response

  1. In lines 125, 153, and 331("or" in eq. 3) there is a problem with font size.

Response: We sincerely thank the reviewer for careful reading and sorry for the errors. These sentences have been revised according to your suggestions.

  1. In lines 158, 214, 215, and 282 you left crossed-out text.

Response: These sentences have been revised according to your suggestions.

  1. In lines 172, 176, and 178 there should be "Schmid".

Response: These errors have been revised.

  1. In line 332 you name Kic as static strain fracture toughness. I think you missed "plane strain" in this name.

Response: We sincerely apologize for this oversight. We have corrected the errors accordingly.

  1. In line 365 Eq. 13 is hardly readable. You can cut the beginning that is taken from the preceding line.

Response: We sincerely thank the reviewer for careful reading. The Eqs.13 and 14 have been revised.

  1. In line 106 you stated that you computed COD using CMOD. Did you assume that the edges of the crack are curvilinear?

Response: The CMOD was used to compute COD in accordance with ASTM standard E647-11 and GB/T 6398-2000. In fact, the shape of crack is not generally considered in COD testing.

  1. When I saw Fig.8 I wondered if you intend to take into account local discrepancies in the state of stress which can lead to these tearing-like voids?

Response: Based on our current understanding, the state of stress of center zone in fracture surfaces should be same.

  1. Did you try to correlate the fracture toughness with the surface parameter? Maybe the surface roughness parameter will be good. Years ago there were attempts to use the fractal dimension of the profile but these were abandoned.

Response: Thanks to the reviewers for the suggestion. We hope to adopt your suggestion in our future research work.
